# Modeling and Predicting the Cell Migration Properties from Scratch Wound Healing Assay on Cisplatin-Resistant Ovarian Cancer Cell Lines Using Artificial Neural Network

**DOI:** 10.3390/healthcare9070911

**Published:** 2021-07-19

**Authors:** Entaz Bahar, Hyonok Yoon

**Affiliations:** Research Institute of Pharmaceutical Sciences, College of Pharmacy, Gyeongsang National University, Jinju 52828, Korea; entazbahar@gnu.ac.kr

**Keywords:** artificial neural network, cell migration assay, scratch wound healing assay, ovarian cancer, cisplatin-resistant

## Abstract

The study of artificial neural networks (ANN) has undergone a tremendous revolution in recent years, boosted by deep learning tools. The presence of a greater number of learning tools and their applications, in particular, favors this revolution. However, there is a significant need to deal with the issue of implementing a systematic method during the development phase of the ANN to increase its performance. A multilayer feedforward neural network (FNN) was proposed in this paper to predict the cell migration assay on cisplatin-sensitive and cisplatin-resistant (CisR) ovarian cancer (OC) cell lines via scratch wound healing assay. An FNN training algorithm model was generated using the MATLAB fitting function in a MATLAB script to accomplish this task. The input parameters were types of cell lines, times, and wound area, and outputs were relative wound area, percentage of wound closure, and wound healing speed. In addition, we tested and compared the initial accuracy of various supervised learning classifier and support vector regression (SVR) algorithms. The proposed ANN model achieved good agreement with the experimental data and minimized error between the estimated and experimental values. The conclusions drawn demonstrate that the developed ANN model is a useful, accurate, fast, and inexpensive method to predict cancerous cell migration characteristics evaluated via scratch wound healing assay.

## 1. Introduction

Cell migration is a vital process in which cells need to adjust and achieve their correct location in a given environment to perform their work [1]. It is possible to deregulate cell migration, which leads to many pathological processes, such as inflammation and cancer metastasis [2,3,4]. Approaches to studying cell migration are particularly interesting in physiology and oncology, as cell migration is relevant to phenotypes when looking into the effects of novel therapeutic drugs and chemoattractants during metastatic progression [5]. The scratch wound healing assay is the most common in vitro biological assay to investigate the mechanisms regulating cancer cell migration or test the efficacy of potential therapeutic drugs. The wound healing assay creates a defined area across which cells migrate. The scratch wound healing assay has been commonly utilized to study the effects of a number of experimental conditions on mammalian cell migration and proliferation, such as gene knockdown or chemical exposure [6]. However, current methods are not efficient enough for in vitro high-throughput screening of small molecules or characterization of the molecular metastatic cascade complex. Most studies on cell migration have the limitation of being based on endpoint assays.

Deep learning-based algorithms, which are based on artificial neural networks (ANN), offer significant promise in extracting features and discovering patterns from large volume data. A highly simplified biological network structure model is the ANN, which learns from examples and recognizes patterns in a series of input and output data without any prior assumptions of their nature and interrelationships. It does not need any mathematical model. The science of neural networks has seen a significant revolution in recent years, aided by deep learning methodologies. The following are some of the advantages of the ANN model: ease of optimization, low cost, flexible nonlinear modeling of large databases, accuracy of predictive implication, and potential to support clinical judgment [7]. By providing an explanation, such as using an extraction rule or sensitivity analysis, the ANN model can make knowledge dissemination easier [8].

Traditional statistical prediction methods such as regression models (e.g., regression splines, projection pursuit regression, penalized regression) involve fitting a model to data, evaluating the fit, and estimating parameters that are later used in a predictive equation [9]. In terms of utility, ANNs are competitive with conventional modeling based on polynomial, linear regression and statistical models [10,11]. Neural networks, discriminant functions, linear classifiers, and support vector classifiers and machines are some other examples of deep learning algorithms. Despite the fact that ANNs provide a more effective, efficient, and successful way to manage both complex and noncomplex data, there is a growing need to address the issue of using a systematic approach during the development phase of ANNs to improve their performance. Many different statistical, probabilistic, and optimization techniques can be implemented including decision trees, discriminant analysis, naïve Bayes, ensemble, support vector machines (SVM), K-nearest neighbor (KNN), and neural network (NN) classifiers, which segment a data set sequentially depending on the correlations between predictor value and an outcome value [12,13]. Meanwhile, a feedforward neural network (FNN) is a machine learning algorithm composed of layers that are relatively simpler to implement and organized similarly to human neuron processing units. There is no feedback between layers when NN operates normally, that is, when it acts as a predictor [14]. It features a straightforward architecture, excellent learning capabilities, and the ability to solve complicated issues. This simple architecture belongs to the shallow network group and is useful for classifying a small number of classes [15]. The ability of the ANN to predict data results in high accuracy of cancer survival prediction [16]. A machine learning-based model identified highly motile cells and nonmotile cells based on microscope image features that determined cell migration ability [17]. Only a few studies have suggested that ANNs achieved optimum accuracy for cell movement direction and speed prediction [18]. However, due to the limitations of conventional marker-based approaches to identify cell migration, we aimed to establish an ANN model to predict cell migration. This paper presents the design, training, and testing of a feedforward ANN to predict the migration capacity of cisplatin-resistant ovarian cancer (OC) cells that makes adequate use of scratch wound healing data from our previously published experimental data.

## 2. Materials and Methods

### 2.1. Assembling Scratch Wound Healing Migration Assay Data for the ANN Model

#### 2.1.1. Cell Culture 

For training the ANN model, we used experimental data from our previous publication [19]. The metastasis properties of the different OC cell lines were determined using a scratch wound healing migration assay. The human serous OC cell line OV-90 and human epithelial OC cell line SKOV-3 were cultured at 37 °C with 5% CO_2_ saturated humidity. The OV-90 was cultivated in a 1:1 mixture of MCDB 105 (LM016-01) and Medium 199 (#GIB-11150-059, Gibco, Life Technologies, Grand Island, NY, USA), while SKOV-3 was cultured in McCoy’s 5A modified (#GIB-16600082, Gibco). All media were supplemented with 10% fetal bovine serum (FBS, #GIB-16000-044, Gibco) and 1% penicillin-streptomycin (#P4333, Sigma Aldrich, St. Louis, MO, USA). We used a constant higher dose (100 µM) of cisplatin for pulse treatment (termed as CisR1), and we started from a lower dose (10, 20, 40, 80, to 100) of cisplatin for intermittent incremental treatment (termed as CisR2) methods to generate CisR OC cells. A total of four sublines were generated from two OC cell lines (OV-90/Parental and SKOV-3/Parental), two from each cell line, including OV-90/CisR1, OV-90/CisR2, SKOV-3/CisR1, and SKOV-3/CisR2.

#### 2.1.2. Scratch Wound Healing Migration Assay

Cell migration was assessed using a scratch wound healing assay following our previous publication [19]. In brief, the parental and CisR cells were cultured in six-well plates for 24 h and then treated with 50 µM cisplatin for another 24 h. Cells were re-suspended and again, 2 × 10^5^ cells were seeded into six-well plates and cultured to monolayers, which were then wounded using sterile 1 mL pipette tips. Cells were washed with PBS to remove any debris. Photos were captured at 0, 12, and 24 h after wounding (Figure 1). The gap distance can be quantitatively evaluated using software such as ImageJ (National Institutes of Health). The equations for calculation of the relative wound area (Equation (1)), percentage (%) of wound closure (Equation (2)), and wound healing speed (Equation (3)) are given below.
(1)$$Relative\;wound\;area=W_{t}/W_{0}$$
(2)$$Wound\;closure\;\left(\%)=((W_{0}-W_{t})/W_{0}\right)\times 100\;$$
(3)$$Healing\;speed\;(µm^{2}/h)=(W_{0}-W_{t})/∆T\;$$*W*_0_ = Wound area at 0 h (µm^2^)*W_t_* = Wound area at ∆h (µm^2^)Δ*T* = Duration of wound measured (h)

### 2.2. Modeling Approach

#### 2.2.1. Automated Analysis by Machine Learning Toolbox

We tested the initial accuracy of various supervised learning algorithm methods, such as decision trees, discriminant analysis, naïve Bayes, ensemble, support vector machines (SVM), K-nearest neighbor (KNN), and neural network (NN) classifiers by using the MATLAB (R2021a) “Classification Learner App”. We applied 5-fold cross-validation to protect against overfitting by partitioning the data set into folds and estimating accuracy on each fold. We used our experimental data set (*n* = 90) for measuring accuracy to select the complex decision tree algorithm. The SVM and NN methods scored the highest accuracy compared with other algorithms (Table 1). The narrow NN algorithm showed the highest accuracy of 86.7% with 5-fold cross validation (Appendix A, Figure A1).

#### 2.2.2. Support Vector Regression (SVR)

Support vector regression (SVR) is an application of the SVM learning algorithm that is highly effective for predicting and recognizing patterns in large numerical datasets [20,21]. We generalized SVR in MATLAB (R2021a) “Regression Learner App” to justify prediction capability of different SVMs, including linear, quadratic, cubic, fine Gaussian, medium Gaussian, and coarse Gaussian SVMs. We could view model statistics in the Current Model Summary pane after training regression models in Regression Learner, and we used these data to assess and compare models. We applied cross-validation to protect against overfitting by partitioning the data set in to folds and estimates accuracy on each fold. We used our experimental data set (*n* = 90) for justifying the prediction capability of different SVMs. We checked the models window after training a model in Regression Learner to find which model had the best overall score. The best RMSE (Validation) is underlined; the root mean square error (RMSE) on the validation set was used to obtain this score (Table 2). This score estimates the trained model’s performance on new testing data.

#### 2.2.3. Multilayer Feedforward Neural Network (FNN)

Computational modeling, mainly using an ANN, can perform precise prediction, processing, and data representation. We utilized an ANN model to predict the migration properties of parental and CisR OC cells based on an in vitro scratch wound healing assay. Figure 2 shows a simplified version of the multilayer feedforward neural network (FNN) model, which includes an input layer, an output layer, and at least one hidden layer in between them [22]. The input layer receives *x* signal externally, and this information is weighted with various synaptic weights (*w_ij_*) and feedforwarded to the hidden layer. Before transmitting the weighted inputs to the output layer, each neuron in the hidden layer integrates them together and applies a nonlinear transfer activation function, *f(a)*.
(4)$$f\left(a\right)=1/\left(1+e^{-a}\right)\;$$

The hidden layer neurons do an identical algorithm with synaptic weight (*w_kj_*) and provide the neural network’s output value, *o*. The hidden and output neurons’ output can be expressed as follows:(5)$$\;p_{j}=f_{h}\;(\underset{i}{\sum }x_{i.}.w_{ji}+b_{j})$$
(6)$$o_{k}=f_{o}\;(\underset{i}{\sum }p_{j}.w_{kj}+b_{k})$$
where *f_h_* and *f_o_* are the activation functions, and *b_j_* and *b_k_* are the bias, of the hidden and output layer, respectively.

The ANN used a feedforward backpropagation model, created using the MATLAB fitting function in a MATLAB script for analyzing the Bayesian regularization algorithm for training and using the mean square error (MSE) method for performance assessment [23,24,25]. Each network was created with three inputs (cell lines, hours, and area) and three outputs (relative wound area, percentage of wound closure, and healing speed). A number of neurons in the hidden layer was selected that could produce better results without overfitting the network. To accomplish this, we trained the network with three input neurons, one hidden layer (~5 to 25 neurons), and three output neurons. For training, validating, and testing the neural networks, the MATLAB script randomly selected 62 (70%) of the samples for the training subset, 14 (15%) for the validation subset and 14 (15%) for the test subset. The obtained error for the proposed ANN model was evaluated using the mean absolute error percentage (MAE%), and the root mean square error (RMSE), and the following equations were used to calculate these data: (7)$$\mathrm{MAE}\%=100\times 1/N\overset{N}{\underset{i=1}{\sum }}[X_{i}\left(\mathrm{Exp}\right)-X_{i}\left(\mathrm{Pred}\right)]$$
(8)$$\mathrm{RMSE}={\left[\frac{\sum_{i=1}^{N}{\left[\;X_{i}\left(\mathrm{Exp}\right)-X_{i}\left(\mathrm{Pred}\right)\right]}^{2}}{N}\right]}^{0.5}$$
(9)$$\mathrm{MSE}=1/N\overset{N}{\underset{i=1}{\sum }}{\left[X_{i}\left(\mathrm{Exp}\right)-{\;X}_{i}\left(\mathrm{Pred}\right)\right]}^{2}$$
where *N* is the number of data and *X*(Exp) and *X*(Pred) are experimental and predicted (ANN) values, respectively.

#### 2.2.4. ANN Modeling via System Identification

Neuromodeling for the prediction of migration ability of cisplatin-resistant ovarian cancer cells was performed via learning of a neural network (NN) [26]. For this, the relative wound area, percentage of wound closure, and wound healing speed were measured using in vitro scratch wound healing assay. Initially, the assay was performed to determine wound area after scratching on two OC cell lines (OV-90/Parental and SKOV-3/Parental), and two sublines from each cell line, including OV-90/CisR1, OV-90/CisR2, SKOV-3/CisR1, and SKOV-3/CisR2, for 0, 12, and 24 h. A total of 92 data points were measured and used to train the NN model. Out of 92 data, 70% of the samples for the training subset, 15% for the validation subset, and 15% for the test subset were randomly chosen. The cell lines, hour (h), and wound area were the primary inputs to NN, and the targets to be learned were the corresponding data of relative wound area, percentage of wound closure, and wound healing speed. The six cell lines, OV-90/Parental, OV-90/CisR1, OV-90/CisR2, SKOV-3/Parental, SKOV-3/CisR1, and SKOV-3/CisR2, were encoded as integers 1, 2, 3, 4, 5, and 6, respectively, and the three time points (0, 12, and 24 h) were encoded as 0, 1, and 2, respectively. The input wound area, output relative wound area, percentage of wound closure, and wound healing speed were expressed with real numbers. As a result, the NN included three input and three output nodes. The number of hidden nodes necessary to learn the system was determined via trial and error. The number of hidden nodes in each hidden layer was steadily increased, starting with five in the first hidden layer. The network was trained on the training dataset for a fixed 1000 epochs using the FNN algorithm with a learning rate of 0.05, and its performance was evaluated by MSE. Table 3 summarizes the findings for different ANN structures. On the training dataset, the networks with eighteen and twenty-one hidden nodes produced the lowest error. Then, although the first hidden layer’s number of nodes was kept at eighteen, a second hidden layer was created to test the network. Starting with two hidden nodes, the second hidden layer’s nodes were gradually increased. The network was then trained and tested as described earlier. Table 4 shows that the NN with ten hidden nodes in the second hidden layer produced the lowest error in both training and testing dataset, but was not superior to the NN with only one hidden layer of eighteen hidden nodes. Therefore, a three-layered NN with three input nodes, eighteen hidden nodes, and three output nodes, namely, a 3-18-3 network, was chosen as optimal network to training and testing wound healing dataset.

## 3. Results

Table 1 depicts the accuracy of different machine learning classifiers algorithms, where SVM and NN classifiers showed the highest scores for classification accuracy. Table 2 shows the performance of different SVR algorithms, where capability of prediction in terms of MAE and RSME was highest in the case of relative wound area, while wound closure and healing speed were very poorly predicted.

Different ANN structures were evaluated and adjusted in this study to find the optimal ANN configuration using the MATLAB (R2019b) NN tool. We tried various architectures with one, two, and three hidden layers, each with a different number of neurons. Table 3 showed the comparison between these structures, where the performance of the NN was expressed as MSE. The Bayesian regularization (BR) algorithm was more efficient than Levenberg–Marquardt optimization and the resilient backpropagation algorithm (RPROP). Table 4 shows that the ANN model with a 3-18-3 structure (e.g., three neurons in the input layer, eighteen neurons in the hidden layer, and three neurons in the output layer) had the lowest MSE. As a result, we chose this structure for our study.

We employed the ANN to create and predict a model in order to determine which factors, including cell lines, hours, and wound area, were most important during cell migration. The performance comparisons of learning rate gradient descent (LEARNGD) versus gradient descent with momentum (LEARNGDM) and activation functions log-sigmoid (Logsig) versus tangent-sigmoid (Transig) are shown in Table 5.

Figure 3 illustrates the proposed ANN model with the network’s performance for testing data versus the number of neurons in the hidden layer using a Bayesian regularization algorithm. After repeated trials, it was found that a network with eighteen neurons in a hidden layer could produce better performance without under- or overfitting (Figure 4). Figure 5 illustrates the MATLAB script of the eighteen neurons, where 70% of the samples for the training subset, 15% for the validation subset, and 15% for the test subset were randomly selected.

Table 6 shows the obtained errors for the proposed ANN model, including the MAE and RMSE values for linear regression between the ANN-predicted and experimental results for the training and testing datasets of each variable.

The testing results for the proposed ANN model in comparison with the experimental results are shown in Table 7. To evaluate the metastasis of acquired CisR OC cells, we generalized an ANN model to predict the migration capability of different sublines based on our in vitro scratch wound healing assays. Metastasis is considered the most critical indicator of cancer recurrence and strongly correlates with a low survival rate [27,28]. Through ANN modeling, we predicted the relative wound area, percentage of wound healing, and wound healing speed at different time points (0, 12, and 24 h) in which higher metastasis of CisR cells was observed. The relative wound area was significantly reduced in CisR cells compared to that in parental OC cells (Appendix A, Figure A2A). The percentage of wound closure and healing speed were higher in CisR OC cells than in parental OC cells (Appendix A, Figure A2B,C). Consistent with our in vitro laboratory results, the ANN learned to predict the migration capability with high accuracy, approximating the experimental data with minimal error.

## 4. Discussion

The purpose of a cancer cell migration monitoring system is to track the metastasis potential of cancer cells so that treatment can be more effective. To be widely used, an experimental technique should not be costly or invasive [29]. For the present study, ANN analysis of cell migration in cisplatin-resistant OC cells was investigated. The MAE and RMSE were used as the error function. To develop an ANN model, the number of layers, the number of neurons in the hidden layer, the learning speeds, and the number of iterations for model training must be carefully considered. For instance, if there are not enough neurons in the hidden layer, the ANN will not detect nonlinear behavior in the training data. On the other hand, if there are too many neurons, the ANN will have an overfitting problem, resulting in a lack of applicability. In this study, we used a trial-and-error approach for this analysis, which is considered the most efficient method for evaluating the required number of neurons, learning rate, and early stopping technique to hinder overfitting [30,31].

In this study, we predicted three parameters of cell migration, including the relative wound area, wound healing capacity, and wound healing speed at 12 and 24 h in four OC cell lines. We found the different extent of migration capacity that represents cancer cell metastasis among these cell types. The CisR cells exhibited higher metastasis ability compared to parental OC cells.

## 5. State of the Art Comparison

Machine learning-based analysis involves the organization and processing of data into input formats that machine learning algorithms can understand. For this, using ImageJ software, the wound area (gap distance) was measured (quantified) from captured images. Then, the machine learning parameters, relative wound area, percentage of wound closure, and healing speed were calculated using Equations (1), (2), and (3), respectively. A dataset (*n* = 90) of six parameters for each cell line was created and used for training of machine learning algorithms.

First, the MATLAB “Classification Learner App” was applied to find the initial accuracy of different classification algorithms, including decision trees (fine, medium, and coarse trees), discriminant analysis (linear and quadratic discriminant), naïve Bayes (Gaussian and Kernel), ensemble (boosted tree, bagged tree, subspace discriminant, subspace KNN, and RUS boosted tree), SVM (linear SVM, quadratic SVM, fine Gaussian SVM, medium SVM, and coarse SVM), KNN (fine KNN, medium, coarse, cosine, cubic, and weighted KNN), and NN (narrow, medium, wider, bilayered, and trilayered NN) classifiers [32]. The predicting accuracy of classifiers was not satisfactory, as presented in Table 1 and Appendix A Figure A1. More effective learning method are needed for predicting experimental data correctly.

Second, the MATLAB “Regression Learner App” was used to compare several SVR algorithms (linear, quadratic, cubic, fine Gaussian, medium Gaussian, and coarse Gaussian) for predicting data. For both training and testing data, all the SVM regression algorithms predicted the relative wound area data with minimum error (MAE and RSME), but poor outcomes were produced in the cases of wound closure and healing speed. In the statistical sciences and the scientific community, classical statistical regression approaches for predictive modeling are accepted but have limited flexibility in the case of a high number of complex datasets. To overcome this limitation, machine learning regression algorithms could be a better approach, but they are also not a full answer because they must be weighed against the limits of the data utilized in the research [33].

Finally, considering all of the above approaches, we implemented an FNN model to predict the data. The MATLAB ANN tool was used to evaluate different network structures and algorithms to optimize configuration for data prediction. The 3-18-3 structure with a BR algorithm effectively predicted both training and testing data. The optimization of FNN architecture is crucial for its better accuracy and faster convergence. For assessing the BR strategy for training and the MSE method for performance assessment, the ANN employed a FNN model developed with the MATLAB fitting function in a MATLAB script. We are convinced that our implementation of MATLAB scripts using FNN is well-suited to match the requirements of the analysis of the migration ability of cisplatin-resistant ovarian cancer cell lines (Table 7, Appendix A Figure A2).

## 6. Conclusions

In conclusion, our ANN model can predict the ability of cisplatin-resistant cancer cells to migrate during the metastasis process. The proposed ANN model obtained good correlation with experimental data with minimum error, and it could do so better than traditional statistical methods or other machine learning algorithms. This approach creates a newer, faster, and more efficient method with a very low cost and high accuracy. Through careful selection of the training algorithm, the ANN predictions for obtaining prognostic information on tumor cell migration capacity were improved. The establishment of this approach could allow researchers to use neural network modeling to identify the best therapeutic efficacy for different cancer cells without having to repeat the process in vitro. However, to determine migratory potential in parental and cisplatin-resistant OC cells via ANN modeling, considerably more research is needed.

## Figures and Tables

**Figure 1 healthcare-09-00911-f001:**
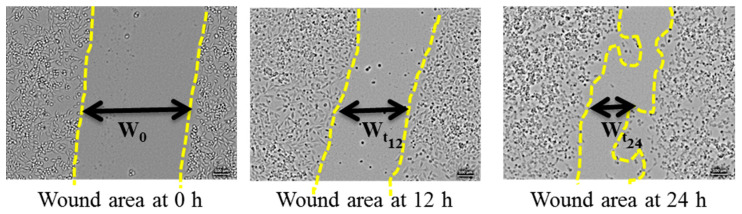
Representative images of scratch wound healing assay at 0, 12, and 24 h.

**Figure 2 healthcare-09-00911-f002:**
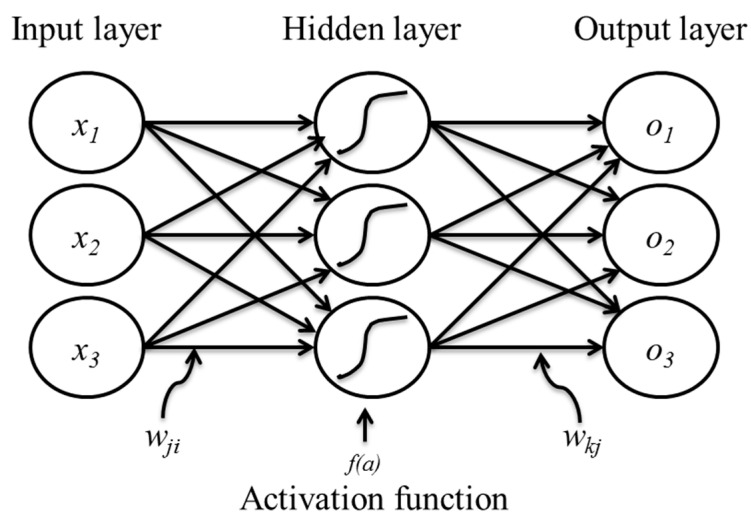
A multilayer feedforward artificial neural network structures with inputs, hidden layer, and outputs.

**Figure 3 healthcare-09-00911-f003:**
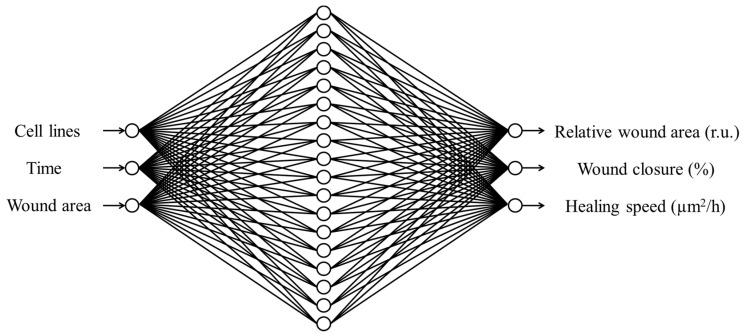
The proposed multilayer feedforward perception (MLP) network consisting of three inputs, one hidden layer with eighteen neurons, and three outputs. r.u.: relative unit, %: percentage; h = hour.

**Figure 4 healthcare-09-00911-f004:**
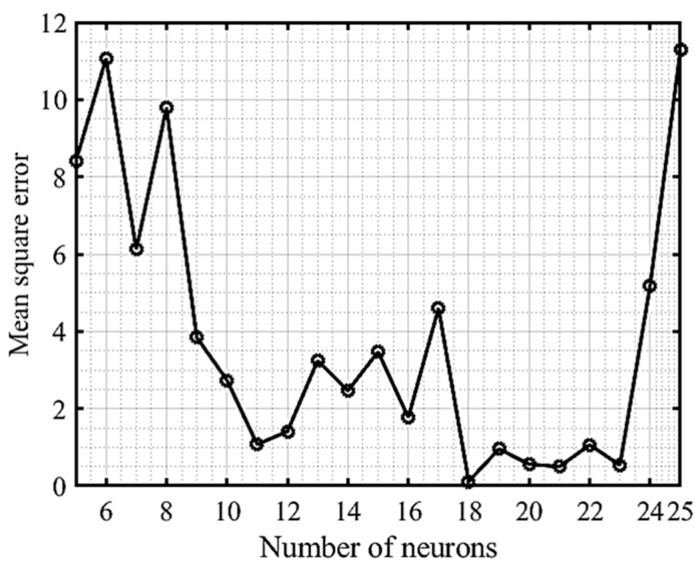
The performance of the network at different hidden neurons using a Bayesian regularization algorithm.

**Figure 5 healthcare-09-00911-f005:**
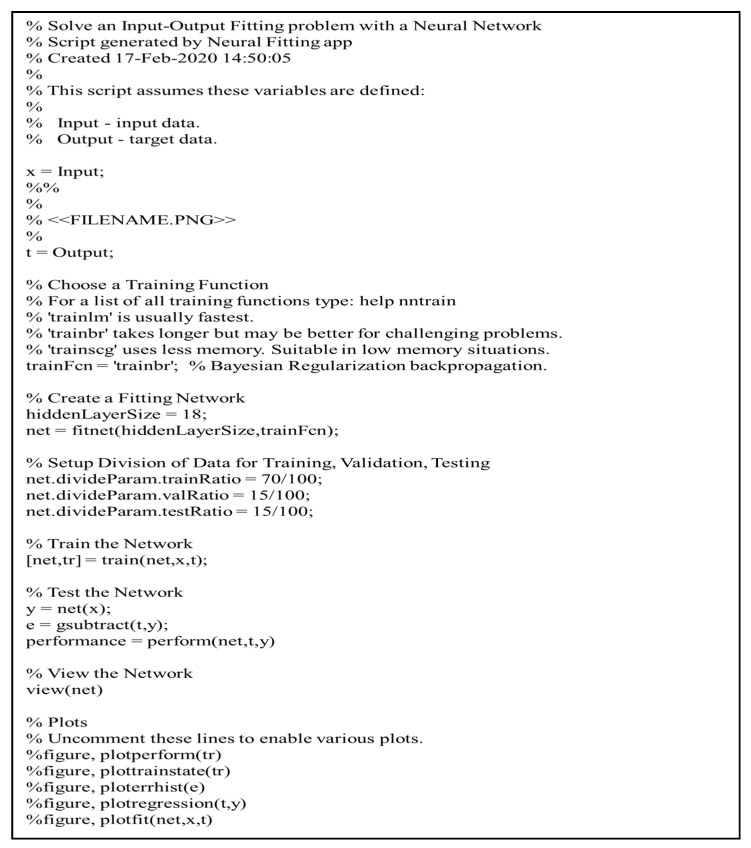
MATLAB script of eighteen neurons.

**Table 1 healthcare-09-00911-t001:** The accuracy of different machines learning algorithms in Classification Learner App.

Classifier		Accuracy (%)	Classifier		Accuracy (%)	Classifier		Accuracy (%)
Decision Tree	Fine Tree	38.9	Support Vector Machines (SVMs)	Linear SVM	67.8	Neural Network (NN)	Narrow NN	86.7
	Medium Tree	38.9		Quadratic SVM	70.0		Medium NN	85.6
	Coarse Tree	24.4		Fine Gaussian SVM	75.6		Wider NN	83.3
Discriminant Analysis	Linear Discriminant	67.8		Medium SVM	70.0		Bilayered NN	82.2
	Quadratic Discriminant	Failed		Coarse SVM	72.2		Trilayered NN	78.9
Naive Bayes	Gaussian	25.6						
	Kernel	27.8	K-Nearest Neighbor (KNN)	Fine KNN	73.3			
Ensemble	Boosted Tree	53.3		Medium KNN	28.9			
	Bagged Tree	57.8		Coarse KNN	16.7			
	Subspace Discriminant	65.6		Cosine KNN	30.0			
	Subspace KNN	42.2		Cubic KNN	31.1			
	RUS Boosted Tree	51.1		Weighted KNN	66.7			

% = percentage; SVM = support vector machine; KNN = K-nearest neighbor; NN = neural network.

**Table 2 healthcare-09-00911-t002:** The performance of different SVM learning algorithms in Regression Learner App.

SVM Machines	Relative Wound Area				Wound Closure				Healing Speed			
	Training		Testing		Training		Testing		Training		Testing	
	MAE	RSME	MAE	RSME	MAE	RSME	MAE	RSME	MAE	RSME	MAE	RSME
Linear SVM	0.028	0.033	0.028	0.033	3.830	5.495	4.204	6.564	12809	16937	1.1 × 10^4^	1.5 × 10^4^
Quadratic SVM	0.028	0.032	0.0278	0.031	3.068	3.586	3.04	4.089	6557.5	8227.3	5.7 × 10^3^	8.1 × 10^3^
Cubic SVM	0.000	0.030	0.0281	0.031	3.491	4.129	3.14	4.309	4052.6	4983	4.1 × 10^3^	4.7 × 10^3^
Fine Gaussian SVM	0.004	0.063	0.0293	0.033	5.208	7.628	1.536	2.553	5949.5	8136.3	4.2 × 10^3^	4.9 × 10^3^
Medium Gaussian SVM	0.027	0.034	0.0262	0.030	2.906	3.610	2.483	3.104	15488	4360.1	3.4 × 10^3^	3.7 × 10^3^
Coarse Gaussian SVM	0.037	0.044	0.0354	0.041	4.281	5.408	4.040	5.275	3676.7	17028	1.1 × 10^4^	1.4 × 10^4^

SVM = support vector machine; MAE = mean absolute error; RMSE = root mean square error.

**Table 3 healthcare-09-00911-t003:** The comparison of different ANN structures’ performance with one, two, and three hidden layers by changing the number of neurons in the hidden layer(s).

ANN Structure	Performance (Average MSE)		
	Training Function Algorithm		
	TrainBR	TrainLIM	TrainRPROP
3-5-3	5.010	9.71 × 10^4^	1.158 × 10^6^
3-7-3	3.140	9.35 × 10^8^	2.01 × 10^6^
3-10-3	2.798	4.21 × 10^8^	3.23 × 10^8^
3-12-3	1.4089	1.75 × 10^6^	7.68 × 10^8^
3-15-3	3.472	8.21 × 10^8^	3.24 × 10^6^
3-18-3	0.058	7.00 × 10^8^	7.32 × 10^6^
3-20-3	0.561	1.82 × 10^6^	7.90 × 10^8^
3-21-3	0.4964	1.65 × 10^6^	7.54 × 10^8^
3-22-3	1.0627	1.25 × 10^6^	7.69 × 10^8^
3-24-3	5.1836	8.65 × 10^4^	1.02 × 10^6^
3-5-5-3	7.460	6.96 × 10^4^	3.21 × 10^6^
3-5-10-3	1.030	1.28 × 10^9^	6.43 × 10^6^
3-10-5-3	0.606	4.78 × 10^3^	1.88 × 10^6^
3-5-5-5-3	70.00	7.53 × 10^4^	7.13 × 10^6^
3-10-5-5-3	1.292	1.66 × 10^5^	2.18 × 10^7^
3-10-10-5-3	1.428	8.38 × 10^4^	6.06 × 10^6^

ANN = artificial neural network; MSE = mean square error; TrainBR = Bayesian regularization; TrainLIM = Levenberg–Marquardt optimization, TrainRPROP = resilient backpropagation algorithm (RPROP).

**Table 4 healthcare-09-00911-t004:** The performance comparison of a two-hidden-layer ANN with various numbers of neurons in the second hidden layer.

ANN Structure	Performance (Average MSE)	
	Training	Testing
3-18-3	0.058	0.012
3-18-2-3	0.460	0.262
3-18-4-3	0.306	0.174
3-18-6-3	0.635	0.345
3-18-8-3	0.752	0.428
3-18-10-3	0.292	0.183
3-18-12-3	1.321	0.752
3-18-15-3	1.024	0.266
3-18-18-3	0.428	0.243
3-18-20-3	0.792	0.451
3-18-22-3	1.428	0.813

ANN **=** artificial neural network; MSE = mean square error.

**Table 5 healthcare-09-00911-t005:** The performance comparison of ANN structure with 3-18-3 based on learning and activation functions.

Neural Network	Adaption Learning Function	Training Function	Activation Function		Performance (MSE)					
					Training			Testing		
			Average (All Outputs)	Relative Wound Area	Wound Closure (%)	Healing Speed (µm^2^/H)	Relative Wound Area	Wound Closure (%)	Healing Speed (µm^2^/H)	
3-18-3	LEARNGD	TrainBR	Logsig	0.0670	0.0036	0.1544	0.4211	0.0029	0.2041	0.5308
			Transig	0.1350	0.0029	0.6259	0.986	0.0041	0.8610	0.5417
	LEARNGDM	TrainBR	Logsig	0.0335	0.0023	0.1113	0.8140	0.0036	0.4536	0.4182
			Transig	0.0319	0.0030	0.1161	0.3014	0.0037	0.6376	0.5299

MSE = mean square error; LEARNGD = learning rate gradient descent; LEARNGDM = learning rate gradient descent with momentum.

**Table 6 healthcare-09-00911-t006:** The average MAE and RSME for outputs for training and testing data.

Output	MAE		RMSE	
	Training	Testing	Training	Testing
Relative wound area	0.0767	0.0528	0.0856	0.0609
Wound closure (%)	0.2090	0.0913	0.2920	0.1220
Healing speed (µm^2^/h)	0.1817	0.0797	0.1060	0.0724

MAE = mean absolute error; RMSE = root mean square error.

**Table 7 healthcare-09-00911-t007:** The comparison between experimental and predicted ANN results for testing data.

		Experiment			ANN		
Cell Lines	Hour	Relative Wound Area (r.u.)	Wound Closure (%)	Healing Speed (µm^2^/H)	Relative Wound Area (r.u.)	Wound Closure (%)	Healing Speed (µm^2^/H)
OV-90/Parental	12	0.752	24.809	43,232.7	0.721	24.877	43,232.5
	24	0.643	35.663	31,073.9	0.527	35.685	31,073.9
OV-90/CisR1	12	0.546	45.388	75,302.5	0.569	45.369	75,302.4
	24	0.365	63.537	52,706.2	0.386	63.685	52,706.2
OV-90/CisR2	12	0.501	49.904	87,289.2	0.529	49.846	87,289.1
	24	0.317	68.321	59,750.6	0.387	68.251	59,750.7
SKOV-3/Parental	12	0.896	10.408	16,036.8	0.845	10.682	16,036.7
	24	0.669	33.145	25,535.0	0.532	33.312	25,534.8
SKOV-3/CisR1	12	0.590	40.954	60,884.5	0.551	41.057	60,884.5
	24	0.296	71.981	56,505.4	0.342	71.885	56,505.4
SKOV-3/CisR2	12	0.551	44.942	70,823.7	0.561	44.915	70,823.6
	24	0.368	74.543	59,811.7	0.360	75.845	59,801.7

CisR1 = cisplatin-resistant subline 1; CisR2 = cisplatin-resistant subline 2.

## Data Availability

The data presented in this study are available in this article Healthcare and its Appendix A.

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
