# Peer review of "Modeling and Predicting the Cell Migration Properties from Scratch Wound Healing Assay on Cisplatin-Resistant Ovarian Cancer Cell Lines Using Artificial Neural Network"

_healthcare, 2021, doi:10.3390/healthcare9070911_

Round 1

Reviewer 1 Report

Dear authors

thank you for the reply. The responses are too concise, could you explain better the concepts?

Best Regards

Author Response

Thank you for the reply. The responses are too concise, could you explain better the concepts?

Reply: We have explained all the sections of manuscript more details to clarify the concepts.

Reviewer 2 Report

I cannot agree that the paper is improved in mentioned points. The paper shows an application of ANN, but the paper still lacks novelty, and the authors do not analyze the latest achievements in machine learning use for modeling purposes (the used bibliography is outdated). The experimental section needs a proper comparison with state-of-art and other tools.

There is still an issue of:
1) The introduction and abstract still are outdated. I cannot agree that the authors improved this section to the current state of knowledge. Analyze mainly papers and research from the last 2 years. Discuss the solutions of using other types of neural networks for similar tasks in medicine or biology for instance analyzing the number of cells, or bacteria on images.
2) Why did you not chose recurrent ANN?
3) There is no proper mathematical model of ANN. How did you chose and define activation function, coefficients in training algorithm, etc.?
4) The authors should justify the choice of ANN and not another tool. 
5) There is no comparison with other solutions like SVM, knn, etc.  It must be added. 
6) Add a comparison with state-of-art.
7) The conclusion is very short and does not say what the proposition was new for this study, there is no information about the advantages/disadvantages of the solutions.

Author Response

 Comments and Suggestions for Authors

I cannot agree that the paper is improved in mentioned points. The paper shows an application of ANN, but the paper still lacks novelty, and the authors do not analyze the latest achievements in machine learning use for modeling purposes (the used bibliography is outdated). The experimental section needs a proper comparison with state-of-art and other tools.

Reply: Thank you very much for your kind comments and suggestions that would help us to improve the quality of our present manuscript as well as future research of AI. 

We have revised our manuscript take all of your following comments into account.

There is still an issue of:

1) The introduction and abstract still are outdated. I cannot agree that the authors improved this section to the current state of knowledge. Analyze mainly papers and research from the last 2 years. Discuss the solutions of using other types of neural networks for similar tasks in medicine or biology for instance analyzing the number of cells, or bacteria on images.

Reply: We have thoroughly revised abstract and introduction based on current state of knowledge.

2) Why did you not chose recurrent ANN?

Reply: Recently, deep learning practitioners have presented plenty of alternative sequence models. Despite recurrent neural networks were once the tool of choice, several researchers have demonstrated that feed-forward networks (FNN) can match the results of the top recurrent models. In addition, Our previous similar type works with ANN given good outcome with FNN.

3) There is no proper mathematical model of ANN. How did you chose and define activation function, coefficients in training algorithm, etc.?

Reply: We have included mathematical function of ANN model in section 2.2.3.

4) The authors should justify the choice of ANN and not another tool.

Reply: We have justified the choice of ANN over other tools.  Please see section 2.2.1, 2.2.2, Table1 and Table 2.

5) There is no comparison with other solutions like SVM, knn, etc.  It must be added.

Reply: We added the comparison of other solutions. Please see section 2.2.1, 2.2.2, Table1 and Table 2.

6) Add a comparison with state-of-art.

Reply: We added a section for comparison with state-of-art. Pleasr see section 5. State of art comparison.

7) The conclusion is very short and does not say what the proposition was new for this study, there is no information about the advantages/disadvantages of the solutions.

Reply: We revised conclusion section with explaining unique finding and merits and demerits of proposed model.

Round 2

Reviewer 2 Report

In my opinion, the paper is still not ready for publication. The novelty is very limited and not new. Moreover, there is no proper analysis of the current knowledge in neural models (the authors used a very simple model), no comparison with state-of-art from the last 4 years.

Author Response

In my opinion, the paper is still not ready for publication. The novelty is very limited and not new. Moreover, there is no proper analysis of the current knowledge in neural models (the authors used a very simple model), no comparison with state-of-art from the last 4 years.

Reply: We have added a section of modeling via system identification to explain the proper analysis of current knowledge in neural network models. Please see section 2.2.3 ANN modeling via system identification.

This manuscript is a resubmission of an earlier submission. The following is a list of the peer review reports and author responses from that submission.

Round 1

Reviewer 1 Report

Dear authors

The manuscript aims to define an algoritm based on artificial intelligence to predict cell migration in ovarian cancer cells resistant to cisplatin.

The referee is an expert in cell biology but not in AI. My comments refer to the bio part.

Major points

It is not clear the number of inputs for the training set. Sixty-two samples refer to what? How many times the data are repeated independently?

Is this prediction specific of migration or is it affected by cell proliferation? Is the rate of cell proliferation included in the analysis?

In the introduction section, there is no analysis of the current status of AI on cell migration. Is this method better than other methods?

Reviewer 2 Report

In the paper, the authors describe the use of ANN for predicting cell migration. In general, the paper has very poor quality and in my opinion, the paper should not be accepted in the current form. The presented results and models are in the initial state of research. Other issues:
The authors do not analyze the current state of knowledge. It is hard to say if the approach is valid and i think it is not - the author uses ANN with very shallow architecture (3 layers), and there is no more test on other architectures. There is an analysis of neurons, layers. I think the recurrent network could be a better method, and some analysis of using it or SVM should be made. Moreover, there is no comparison with other state-of-art methods. The used parameter in the network also is very unknown for readers. 
The paper has no novelty and a very basic solution. Therefore i recommend rejecting it.